# #ForYou? the impact of pro-ana TikTok content on body image dissatisfaction and internalisation of societal beauty standards

**Madison R. Blackburn, Rachel C. Hogg** *

Faculty of Business, School of Psychology, Justice and Behavioural Science, Charles Sturt University, Wagga Wagga, New South Wales, Australia

* rhogg@csu.edu.au

## Abstract

Videos glamourising disordered eating practices and body image concerns readily circulate on TikTok. Minimal empirical research has investigated the impact of TikTok content on body image and eating behaviour. The present study aimed to fill this gap in current research by examining the influence of pro-anorexia TikTok content on young women's body image and degree of internalisation of beauty standards, whilst also exploring the impact of daily time spent on TikTok and the development of disordered eating behaviours. An experimental and cross-sectional design was used to explore body image and internalisation of beauty standards in relation to pro-anorexia TikTok content. Time spent on TikTok was examined in relation to the risk of developing orthorexia nervosa. A sample of 273 female-identifying persons aged 18–28 years were exposed to either pro-anorexia or neutral TikTok content. Pre- and post-test measures of body image and internalisation of beauty standards were obtained. Participants were divided into four groups based on average daily time spent on TikTok. Women exposed to pro-anorexia content displayed the greatest decrease in body image satisfaction and an increase in internalisation of societal beauty standards. Women exposed to neutral content also reported a decrease in body image satisfaction. Participants categorised as high and extreme daily TikTok users reported greater average disordered eating behaviour on the EAT-26 than participants with low and moderate use, however this finding was not statistically significant in relation to orthorexic behaviours. This research has implications for the mental health of young female TikTok users, with exposure to pro-anorexia content having immediate consequences for internalisation and body image dissatisfaction, potentially increasing one's risk of developing disordered eating beliefs and behaviours.

## Introduction

Social media is a self-presentation device, a mode of entertainment, and a means of connecting with others [1], allowing for performance and the performance of identity [2], with social rewards built into its systems. Five to six years of the average human lifespan are now spent on

**Data Availability Statement:** The data for this study can be found on Figshare via the following link: https://doi.org/10.6084/m9.figshare.25756800.v1.

**Funding:** We acknowledge the financial support provided by Charles Sturt University.

**Competing interests:** The authors have declared that no competing interests exist.

social media sites [3] and visual platforms such as Instagram and TikTok increasingly dominate the cultural landscape of social media. Such visually oriented platforms are associated with higher levels of dysfunction in body image [4], while the COVID-19 pandemic has seen a rise in disordered eating behaviour [5]. Despite this, the field lacks a clear theoretical framework for understanding how social media usage heightens body image issues [6] and little research has specifically examined the impacts of TikTok based content. In this research, we sought to explore the impact of pro-anorexia TikTok content on body image satisfaction and internalisation of beauty standards for young women. The forthcoming sections of this literature review will highlight the features of social media content that may be particularly pernicious for young female users and will explore disordered eating and orthorexia in a social media context, concluding with a theoretical analysis of the relationship between social media and body image and internalisation of beauty standards, respectively.

Social media offers instant, quantifiable feedback coupled with idealised online imagery that may intersect with the value adolescents attribute to peer relationships and the sociocultural gender socialisation processes germane to this period of development, creating the "perfect storm" for young social media users, especially females [6]. In a study of 85 young, largely female eating disorder patients, a rise in awareness of online sites emphasizing thinness as beauty was evident from 2017 to 2020, with 60% of participants indicating that they knew of pro-ana websites and 22% of participants admitting to visiting them [7]. Research suggests that social media may also trigger those with extant eating disorders while simultaneously influencing healthy individuals to engage in disordered eating behaviour [8].

"Pro" eating disorder communities, hereafter referred to as "pro-ana" (pro-anorexia) communities, are a particular concern in a social media context. These communities encourage disordered eating, normalise disordered behaviours, and provide a means of connection for individuals who endorse anti-recovery from eating disorders [8]. Weight-loss tips, excessive exercise routines, and images of emaciated figures are routinely shared in these online communities [9], with extant research highlighting the association between viewing eating disorder content online and offline eating disorder behaviour [8]. Women who view pro-ana websites display increased eating disturbances, lowered body satisfaction, an increased drive for thinness, and higher levels of perfectionism when compared to women who have not viewed pro-ana content [10,11]. In research on adolescent girls, Stice [12] investigated the influence of exposure to media portraying the "thin-ideal" and found that perceived pressure to be thin was a predictor of increased body image dissatisfaction, which in turn led to increases in disordered eating behaviour. In similar research, Green [10] found that individuals with diagnosed eating disorders reported worsening symptoms after just 10-minutes of exposure to pro-ana content on the online platform, Tumblr.

## Disordered eating #ForYou

The most downloaded social application (app) of 2021, TikTok is a social media platform that allows short-form video creation and sharing within a social media context [13]. Since its launch in 2017, TikTok has had over two billion downloads and has an estimated one billion users, the vast majority of which are children and teenagers [14]. Unlike other social media platforms where users have greater autonomy over the content generated on their homepage newsfeed, TikTok's algorithm records data from single users and proposes videos designed to catch a user's attention specifically, by creating a personalised "For You" page [15]. This feed will suggest videos from any creator on the platform, not just followed accounts. As such, if a user 'interacts' with a video, such as liking, sharing, commenting, or searching for related content, the algorithm will continue to produce similar videos on their "For You" page. The speed

with which TikTok content can be created and consumed online may also be key to its impact. Any given social media user could watch more than a thousand videos on TikTok in an hour, creating a reinforcing effect that may have more impact than longer form content from a single creator [2].

Whilst the popularity of TikTok's "For You" page has prompted global leaders in social media to build their own recommended content features, this feature remains most pronounced on TikTok. The "For You" page is the homepage of TikTok where users spend the majority of their time, compared to other social media platforms where homepages consist of a curation of content from followed accounts. Instagram's explore page continues to emphasise established influencer culture and promote accounts of public figures or influencers with large followings. Contrastingly, TikTok's unique algorithm makes content discoverability an even playing field, as any user's content has the potential to reach a vast audience regardless of follower count or celebrity status. TikTok users therefore have less control over their homepage newsfeed compared to other social media platforms where users elect who they follow.

Unlike other social media platforms that implicitly showcase body ideals, TikTok contains explicit eating disorder content [16], while the "For You" page means that simply interacting with health and fitness videos can lead to unintended exposure to disordered eating content. Even seemingly benign "fitspiration" content may have psychological consequences for viewers. Beyond explicit pro-ana content, #GymTok and #FoodTok are two popular areas of content that provide a forum for users to create and consume content around their and others' daily eating routines, weight loss transformations, and workout routines [2]. TikTok also frequently features content promoting clean eating, detox cleanses, and limited ingredient diets reflective of the current "food as medicine" movement of western culture [17], otherwise known as orthorexia. Despite efforts to ban such pro-ana related content, some videos easily circumvent controls [18], in part because many TikTok creators are non-public figures who are not liable to the backlash or cancellation that a public figure might receive for circulating socially irresponsible content.

## Orthorexia: The rise of 'healthy' eating pathologies

Psychological analyses of eating disorders have historically focused on restrictive eating and the binge-purge cycle, however, more recently "positive" interests in nutrition have been examined. Orthorexia nervosa is characterised by a restrictive diet, ritualized patterns of eating, and rigid avoidance of foods deemed unhealthy or impure that consumes an individual's focus [19]. Despite frequent observation of this distinct behavioural pattern by clinicians, orthorexia has received limited empirical attention and is not formally recognised as a psychiatric disorder [19]. Orthorexia and anorexia nervosa share traits of perfectionism, high trait anxiety, a high need to exert control, plus the potential for significant weight loss [19]. Termed 'the disorder that cannot be diagnosed' due to limited consensus around its features and the line between healthy and pathological eating practices, orthorexia mirrors the narrative of neoliberal self-improvement culture, wherein the body is treated as a site of performance and transformation.

Orthorexic restrictions and obsessions are routinely interpreted as signs of morality, health consciousness, and wellness [20,21]. Social media wellness influencers have played a significant role in normalising "clean [disordered] eating". As one example, Turner and Lefevre [22] conducted an online survey of social media users following health food accounts and found that higher Instagram use was associated with a greater tendency towards orthorexia, with the prevalence of orthorexia among the study population at 49%, substantially higher than the general population (<1%). Similar health and food-related content on TikTok may provoke orthorexic

tendencies among TikTok users, however, limited research has investigated orthorexic eating behaviour in the context of TikTok. The current study aims to bridge this gap in the literature around TikTok use and orthorexic tendencies. Disordered eating behaviour in the present study was measured by two separate but related constructs. 'Restrictive' disordered eating relates to dieting, oral control, and bulimic symptoms, whilst 'healthy' disordered eating constitutes orthorexic-like preoccupation with health food.

## Theoretical analysis of body image and social media

An established risk factor in the development and maintenance of disordered eating behaviour is negative body image. Body image is a multidimensional construct that represents an individual's perceptions and attitudes about their physical-self and encompasses an evaluative function through which individuals compare perceptions of their actual "self" to "ideal" images [23]. This comparison may produce feelings of dissatisfaction about one's own body image if a significant discrepancy exists between the actual and ideal self-image [23]. Body image is not necessarily congruent with actual physique, with research demonstrating that women categorised as having a healthy body mass index (BMI) nonetheless report dissatisfaction with their weight and engage in restrictive dietary behaviours to reduce their weight [24]. In addition, body image dissatisfaction is considered normative in Western society, particularly among adolescent women [25]. This may be attributable to the constant flow of media that exposes women to unrealistic images of thinness idealized within society [26].

One theoretical framework for understanding social media's relationship with body image is the Social Comparison Theory, proposed by Festinger [27] who suggests that people naturally evaluate themselves in comparison to others via upward or downward social comparisons. Research supports the notion that women who frequently engage in maladaptive upward appearance-related social comparisons are more likely to experience body image dissatisfaction and disordered eating [25,28], while visual exposure to thin bodies may detrimentally modulate one's level of body image satisfaction [29–31]. In their study of undergraduate females, Engeln-Maddox [29] found that participants made upward social comparisons to images of thin models which were strongly associated with decreases in body image satisfaction and internalisation of thinness. Similarly, Tiggemann [32] found that adolescents who spent more time watching television featuring attractive actors and actresses reported an increased desire for thinness, theorised to be a result of increased social comparison to attractive media personalities.

The Transactional Model [33] extends Social Comparison Theory by emphasising the multifaceted and complex nature of social media influences on body image. This model acknowledges that individual differences may predispose a person to utilise social media for gratification, and highlights that as time spent on social media increases, so too does body image dissatisfaction [33]. In line with this, a recent review of literature by Frieiro Padín and colleagues [34] indicated that time spent on social media was strongly correlated with eating disorder psychopathologies, as well as heightened body image concerns, internalisation of the thin ideal, and lower levels of self-esteem. Time on social media also correlated with heightened body image concerns to a far greater extent than general internet usage [35,36].

Body image ideals are not static. The traditional ideal of rib-protruding bodies from the 90s, known colloquially as "heroin chic", have recently shifted to a celebration of the "slim-thicc" figure, consisting of a cinched, flat waist with curvy hips, ample breasts, and large behinds [37]. The "slim-thicc" aesthetic allows women to be bigger than previous body ideals, yet this figure is arguably more unattainable than the thin-ideal as surgical intervention is commonly needed to achieve it, depending on genetics and body type. The idealisation of the

"slim-thicc" figure is highlighted by the "Brazilian butt lift" (BBL), a potentially life-threatening procedure that is nonetheless the fastest growing category of plastic surgery, doubling in growth over the past five years, despite the life-threatening potential of the procedure [38]. Research suggests that the slim-thicc ideal is no less damaging nor threatening of body image than the thin-ideal. Indeed, in experimental research on body ideals, McComb and Mills [39] found that the greatest body dissatisfaction levels in female undergraduate students were observed among those exposed to imagery of the slim-thicc physique, relative to that exhibited by those exposed to the thin-ideal and fit-ideal physique, as well as the control condition.

Recent body ideals have also favoured muscular thin presentations, considered to represent health and fitness as evident in the "#fitspiration" Instagram hashtag that features over 65 million images [40]. Fitspiration has the potential to positively influence women's health and well-being by promoting exercise engagement and healthy eating, yet various content analyses of fitspiration images highlight aspects of fitspiration that warrant concern [see 40,41]. Notably, fitspiration typically showcases only one body type and women whose bodies do not meet this standard may experience body dissatisfaction [40], while the gamification of exercise, such as receiving likes for every ten sit-ups, segues with the intensive self-control and competitiveness that often underpins eating disorders and eating disorder communities [1].

In recent experimental research, Pryde and Prichard [42] examined the effect of exposure to fitspiration TikTok content on the body dissatisfaction, appearance comparison, and mood of young Australian women. Viewing fitspiration TikTok videos led to increased negative mood and increased appearance comparison but did not impact body dissatisfaction. This finding contradicts previous research and may be due to fitspiration content showcasing body functionality rather than aesthetic, which may lead to positive outcomes for viewers. The fitspiration content used by Pryde and Prichard [42] did not contain the harmful themes regularly found in other forms of fitspiration content. Appearance comparison was significant in the relationship between TikTok content and body dissatisfaction and mood, suggesting that this may be a key mechanism through which fitspiration content leads to negative body image outcomes and supporting the notion that fitspiration promotes a focus on appearance rather than health.

Body image dissatisfaction among women is associated with co-morbid psychological disturbances and the development of disordered eating behaviours [43,44]. A large body of research indicates that higher levels of both general and appearance-related social comparison are associated with disordered eating in undergraduate populations [10,28,45–48]. As one example, Lindner et al. [46] investigated the impact of the female-to-male ratio of college campuses on female students' engagement in social comparison and eating pathology. Their findings lend support to the Social Comparison Theory, indicating that the highest levels of eating pathology and social comparison were found among women attending colleges with predominantly female undergraduate populations. A strong relationship was also found between eating pathology and engagement in appearance-related social comparisons independent of actual weight. Lindner et al. [46] surmised that these results suggest social comparison and eating pathology behaviours are due to students' perceptual distortions of their own bodies, potentially fostered by pressures exerted from peers to be thin.

Similarly, Corning et al. [45] investigated the social comparison behaviours of women with eating disorder symptoms and their asymptomatic peers. Results illustrated that a greater tendency to engage in everyday social comparison predicted the presence of eating disorder symptoms, while women with eating disorder symptoms made significantly more social comparisons of their own bodies. Such findings are supported by subsequent research, with Hamel et al. [28] finding that adolescents with a diagnosed eating disorder engaged in significantly more body-related social comparison than adolescents diagnosed with a depressive disorder

or no diagnosis. Body-related social comparison was also significantly positively correlated with disordered eating behaviours. While extant research has focused upon social comparison as it has occurred through traditional media outlets, less research has investigated the facilitation of social comparison through social media platforms, particularly contemporary platforms such as TikTok.

## Theoretical analysis of internalisation processes and social media

The extent to which one's body image is impacted by images and messages conveyed by the media is determined by the degree to which these images and messages are internalised. Some may argue that social media platforms are distinct from what occurs in "real" life, creating fewer opportunities for internalisation to occur. Yet as Pierce [2] argues, platforms such as Tik-Tok create their own realities, allowing users to explore their identities, form relationships, engage with culture and world events, and even develop new patterns of speech and writing. TikTok trends commonly infiltrate society, underscoring the impact of social media on life beyond the online world and thus a sociocultural analysis of TikTok is warranted. Sociocultural theories suggest that society portrays thinness as the ideal body shape for women, resulting in an internationalisation of the "thin is good" assumption for women. This in turn results in lowered body image satisfaction and other negative outcomes [43]. The significance of social influences, including the role of family, peers, and the media, is emphasised by sociocultural theory, with individuals more likely to internalise the thin ideal when they encounter pressuring messages that they are not thin enough from social influences [48]. Within this theoretical approach, an individual's degree of thin ideal internalisation is theorised to depend on their acceptance of socially defined ideals of attractiveness and is reflected in their engagement in behaviours that adhere to these socially defined ideals [49].

Building on this, the tripartite influence model suggests that disordered eating behaviours arise due to pressure from social agents, specifically media, family, and peers. This pressure centres on conforming to appearance ideals and may lead to engagement in social comparison and the internalisation of thin ideals [48]. This is relevant in a digital context given social media provides endless opportunities for individuals to practice social comparison and for many users, social comparison on TikTok is peer-based as well as media-based. According to the tripartite model, social comparisons have been consistently associated with a higher degree of thin ideal internalisation, self-objectification, drive for thinness, and weight dissatisfaction [50]. Furthermore, and in contrast to traditional media where social agents are mainly models, celebrities, and movie stars, social agents on social media can include peers, friends, family, and individuals who have a relationship with the individual. Social media content generated by "everyday" people, rather than super models or movie stars, may result in comparisons that are more horizontal in nature. This is particularly evident on TikTok where content creators are rarely famous before creating a TikTok account, often remain micro-influencers after achieving some notoriety, and are usually around the same age as those viewing their content.

Pressure to be thin from alike peers may have a particularly pronounced impact on one's degree of internalisation of the thinness ideal. Indeed, Stice et al. [51] found that after listening to young thin women complain about "feeling fat", their adolescent participant sample reported increased body image dissatisfaction, suggesting that pressure from peers perpetuates the thinness ideal, leading to internalisation of the ideal and subsequent body dissatisfaction. Similarly, it was found that adolescent females were more likely to engage in weight loss behaviour if a high portion of peers with a similar BMI were also engaging in these behaviours, illustrating that appearance pressure exerted by alike peers may result in thin-ideal internalisation and engagement in weight loss behaviours to control body weight and shape [52]. Such

findings raise questions around whether those most similar to us have the greatest impact upon thin-ideal internalisation, body image dissatisfaction, and disordered eating behaviours.

In further support for the tripartite influence model, research by Thompson et al. [48] indicates that the ideals promoted through social media trends are internalized despite being unattainable, resulting in body image dissatisfaction and disordered eating behaviour. Similarly, Mingoia et al. [53] found a positive association between the use of social networking sites and thin ideal internalisation in women, indicating that greater use of social networking sites was linked to significantly higher internalisation of the thin ideal. Interestingly, the use of appearance-related features (e.g., posting or viewing photographs or videos) was more strongly related with internalisation than the broad use of social networking sites (e.g., writing status', messaging features) [53]. Correlational and experimental research alike has demonstrated that thin ideal internalisation is related to body image dissatisfaction and leads to expressions of disordered eating such as restrictive dieting and binge-purge symptoms [31,48,54,55]. Subsequent expressions of disordered eating may be seen as an attempt to control weight and body shape to conform to societal beauty standards of thinness [51].

This sociocultural perspective is exemplified by Grabe et al's. [54] meta-analysis of research on the associations between media exposure to women's body dissatisfaction, internalisation of the thin ideal, and eating behaviours and beliefs, illustrating that exposure to media images propagating the thin ideal is related to and indeed, may lead to body image concerns and increased endorsement of disordered eating behaviours in women. Similarly, Groesz et al. [55] conducted a meta-analysis to examine experimental manipulations of the thin beauty ideal. They found that body image was significantly more negative after viewing thin media images than after viewing images of average size models, plus size models, or inanimate objects. This effect size was stronger for participants who were more vulnerable to activation of the thinness schema. Groesz et al. [55] conclude that their results align with the sociocultural theory perspective that media promulgates a thin ideal that in turn provokes body dissatisfaction.

## Current research

Existing research has established the relationship between body image dissatisfaction and disordered eating behaviours and social media platforms such as Instagram and Twitter. The unique implications of the TikTok 'For You Page', as well as the dominance of peer-created and explicit disordered eating content on TikTok suggests that the influence of this platform warrants specific consideration. This study adds to extant literature by utilising an experimental design to examine the influence of exposure to pro-ana TikTok content on body image and internalisation of societal beauty standards. A cross-sectional design was used to investigate the effect of daily TikTok and the development of disordered eating behaviours. Although body image disturbance and eating disorders are not limited to women, varying sociocultural factors have been implicated in the development of disordered eating behaviour in men and women [45], while issues facing trans people warrant specific consideration beyond the scope of this study, therefore the present sample contains only female-identifying participants.

**Aims and hypotheses.** The current study aimed to investigate the impact of pro-ana TikTok content on young women's body image satisfaction and internalisation of beauty standards, as well as exploring daily TikTok use and the development of disordered eating behaviour. First, in line with the cross-sectional component of the study, it was hypothesized that women who spend greater time on TikTok per day would report significantly more disordered eating behaviour than women who spend low amounts of time on TikTok per day. Second, it was hypothesized that women in the pro-ana TikTok group would report a significant decrease in body image satisfaction state following exposure to the pro-ana content compared

to women in the control group. Third, it was hypothesized that women in the pro-ana Tik Tok group would report increased internalisation of societal beauty standards following exposure to pro-ana TikTok content compared to women in the control group.

## Method

### Participants

Participants in the current study included 273 women aged between 18 to 28 years sourced from the general population of TikTok users. The predominant country of residence of the sample was Australia, with 15 participants indicating they currently reside outside of Australia. Of the remaining data relating to the two conditions of the study, 126 participants were randomly allocated into the experimental condition, and 147 participants were randomly allocated into the control condition. Snowball sampling was used to recruit participants through social media, online survey sharing platforms, and word-of-mouth, with first-year University students targeted for recruitment by offering class credit in return for participation. Participants could withdraw their consent at any time by exiting the study prior to completion of the survey.

### Measures

The current study employed a questionnaire set that included a demographic questionnaire, and five scales measuring disordered eating behaviour, body satisfaction, and internalisation of societal beauty standards, as well as perfectionism, the latter of which was not examined in the present study.

**Demographic questionnaire.** The demographic questionnaire required participants to answer a series of questions relating to their gender, age, relationship status, ethnicity, country of residence, TikTok usage, and exercise routine. A screening question redirected non-female-identifying persons from the study. Responses to the TikTok usage items were examined cross-sectionally with responses on the EAT-26 and ORTO15 used to examine the influence of daily TikTok use and the presentation of disordered eating behaviours.

**Eating attitudes test.** The Eating Attitudes Test (EAT-26, [56]) is a short form of the original 40-item EAT scale [57] which measures symptoms and concerns characteristic of eating disorders. The 26-item short-form version of the EAT was utilised in the present study due to its established reliability and validity, and strong correlation with the EAT-40 [56].

Responses to the 26-items are self-reported using a 6-point Likert scale ranging from *Always* (3) to *Never* (0) [56]. The EAT-26 consists of three subscales including dieting, bulimia and food preoccupation, and oral control. Five behavioural questions are included in Part C of the EAT-26 to determine the presence and frequency of extreme weight-control behaviours including binge eating, self-induced vomiting, laxative usage, and excessive exercise [56]. Higher scores indicate greater disordered eating behaviour, and those with a total score of 20 or greater are, in clinical contexts, typically highlighted as requiring further assessment and advice of a mental health professional [56].

Internal consistency of the EAT-26 was established in initial psychometric studies which reported a Cronbach's alpha of.85 [58]. For the current study, the Cronbach's alpha = .91. Previous research has also demonstrated that the EAT-26 has strong test-retest reliability (e.g., 0.84) [59], as well as acceptable criterion-related validity for differentiating between eating disorder populations and non-disordered populations [56]. In the current study, the EAT-26 was used to measure disordered eating behaviour, and the cut-off score of 20 and above was adopted to categorise increased disordered eating behaviour. Given how this construct is

measured, from this point forward the present study will refer to EAT-26 responses as 'restrictive' type disordered eating.

**ORTO-15.** The ORTO-15 is a 15-item screening measure that assesses orthorexia nervosa risk through questions regarding the perceived effects of eating healthy food (e.g. "Do you think that consuming healthy food may improve your appearance?"), eating habits (e.g. "At present, are you alone when having meals?"), and the extent to which concerns about food influence daily life (e.g. "Does the thought of food worry you for more than three hours a day?") [19]. Responses are self-reported using a 4-point Likert scale ranging from *always*, *often*, *sometimes*, or *never*. Individual items are coded and summed to derive a total score. Donini et al. [60] established a cut off total score of 40; scores below 40 indicate orthorexia behaviours, whilst scores 40 or above reflect normal eating behaviour. This cut off score was determined by Donini et al. [60] as their results revealed the ORTO-15 demonstrated good predictive capability at the threshold of 40 compared to other potential threshold values.

Although the ORTO-15 is the most widely accepted screening tool to assess orthorexia risk, it is still only partially validated [61], and inconsistencies of the measures' reliability and validity exist in current literature. For example, Roncero et al. [62] estimated that the reliability of the ORTO-15 using Cronbach's alpha was between 0.20 and 0.23, however, after removing certain items, the reliability coefficients were between 0.74 and 0.83. Contrastingly, Costa and colleagues' [63] review of current literature surrounding orthorexia suggested adequate internal consistency (Cronbach's alpha = 0.83 to 0.91) with all 15-items.

In the present study, a reliability analysis revealed unacceptable reliability for the ORTO-15 ($\alpha$ = .24). Principal components factor analysis identified two factors within the ORTO-15, one relating to dieting and the other to preoccupation with health food. Separate reliability analyses were performed on the items that comprised these two factors and the diet-related items did not have acceptable reliability ($\alpha$ = -.40), whilst the health food-related items bordered on acceptable reliability at $\alpha$ = .63. Consequently, only the health food-related items were retained in the current study following consideration of Pallant's [64] assertion that Cronbach alpha values are sensitive to the number of items on a scale and it is therefore common to obtain low values on scales with less than ten items. Pallant [64] notes that in cases such as this, it is appropriate to report the inter-item correlation of the items, while Briggs and Cheek [65] advise an optimal range for the inter-item correlation between.2 to.4, with the health food-related items in the current study obtaining an inter-item correlation of.25. Throughout this study, the construct measured by these ORTO-15 items will be referred to as 'healthy' type disordered eating to reflect this obsessive health food preoccupation and differentiate between the two disordered eating dependent variables measured in the current study.

**Body image states scale.** The Body Image States Scale (BISS) by Cash and colleagues [66] is a six-item measure of momentary evaluative and affective experiences of one's own physical appearance. The BISS evaluates the following aspects of current body experience: dissatisfaction-satisfaction with overall physical appearance; dissatisfaction-satisfaction with one's body size and shape; dissatisfaction-satisfaction with one's weight; feelings of physical attractiveness-unattractiveness; current feelings about one's looks relative to how one usually feels; and evaluation of one's appearance relative to how the average person looks [66]. Participants responded to these items using a 9-point Likert-type scale which is presented in a negative-to-positive direction for half of the items, and a positive-to-negative direction for the other half [66]. Respondents were instructed to select the statement that best captured how they felt "*right now at this very moment*". A total BISS score was calculated by reverse-scoring the three positive-to-negative items, summing the six-items, and finding the mean, with higher total BISS scores indicating more favourable body image states.

During the development and implementation of the BISS, Cash and colleagues [66] report acceptable internal consistency and moderate stability over time, an anticipated outcome due to the nature of the BISS as a state assessment tool. The BISS was also appropriately correlated with a range of trait measures of body image, highlighting its convergent validity [66]. Cash and colleagues [66] also report that the BISS is sensitive to reactions in positive and negative situational contexts and has good construct validity. An acceptable Cronbach's alpha coefficient of.88 was obtained in the current study.

**Sociocultural Attitudes Towards Appearance Questionnaire—4.** The Sociocultural Attitudes Towards Appearance Questionnaire– 4 (SATAQ-4) [67] is a 22-item self-report questionnaire that assesses the influence of interpersonal and sociocultural appearance ideals on one's body image, eating disturbance, and self-esteem. Ratings are captured on a 5-point Likert scale which asks participants to specify their level of agreement with each statement by choosing from 1 (*definitely disagree*) through to 5 (*definitely agree*), with higher scores indicative of greater pressure to conform to, or greater internalisation of, interpersonal and sociocultural appearance ideals [67]. The five subscales of the SATAQ-4 measure: internalisation of thin/low body fat ideals, internalisation of muscular/athletic ideals, influence of pressures from family, influence of pressure from peers, and influence of pressures from the media [67]. For the purposes of the present study, the questions from the media pressure subscale were modified to enquire specifically about social media rather than traditional forms of media.

Across all samples in Schaefer et al's. [67] study, the internal consistency of the five SATAQ-4 subscales is considered acceptable to excellent, with Cronbach's alpha scores between 0.75 and 0.95. These subscales also displayed good convergent validity with other measures of body satisfaction, eating disorder risk, and self-esteem [67]. Pearson product-moment correlations between the SATAQ-4 subscales and convergent measures revealed medium to large positive associations with eating disorder symptomology, medium negative associations with body satisfaction, and small negative associations with self-esteem [67]. A Cronbach's alpha of.87 was obtained in the present study, demonstrating acceptable internal consistency.

## Procedure

Ethical approval for the present study was granted by the Charles Sturt University Human Research Ethics Committee (Approval number H21155) prior to data collection. Participants were directed to the study via an online link to QuestionPro where they were provided an explanation of the study, their rights, and contact details of relevant support services if they were to become distressed. Participants gave informed consent by clicking on a link that read, "I consent to participate" at the beginning of the survey and then again through the submission of their completed survey. Any incomplete responses were not included in the dataset. Data collection commenced on the 30[th] of July 2021 and ceased on the 1st of October 2021. In line with the cross-sectional and descriptive aspects of the research, participants were asked demographic questions about their gender, age, relationship status, ethnicity, country of residence, TikTok usage, and exercise habits. Participants then completed the experimental set in the following order: BISS (pre-test), SATAQ-4 (pre-test), EAT-26, ORTO-15, Experimental intervention (control or experimental TikTok video condition), SATAQ-4 (post-test), BISS (post-test), and debrief. All questionnaires presented to each participant were identical. Measures were not randomised to ensure that body image and internalisation were assessed at both pre- and post-test to evaluate the experimental manipulation.

Participants were randomly allocated to one of two conditions: experimental (pro-ana TikTok video) or control ("normal" TikTok video). Participants allocated to the experimental

condition watched a compilation of TikTok videos containing explicit disordered eating messages such as young women restricting their food, displaying gallows humour about their disordered eating behaviour, starving themselves, and providing weight loss tips such as eating ice cubes and chewing gum to curve hunger. Participants in the experimental condition were also exposed to more implicit body image ideals typical of fitspiration-style content. This included thin women displaying their abdomens, cinched waists, dancing in two-piece swimwear, along with workout and juice cleanse videos promising fast weight loss. Participants in the control condition viewed a compilation of TikTok videos containing scenes relating to nature, cooking and recipes, animals, and comedy. After viewing the 7- to 8-minute TikTok video, all participants completed measures of internalisation and body satisfaction again to assess the influence of either the pro-ana TikTok video or the normal TikTok video. The debrief statement made explicit to participants the rationale of the study and explained the non-normative content of the videos shown to the experimental group. A small financial incentive was offered via a prize draw of five vouchers.

## Statistical analysis

The data from QuestionPro was collated and analysed using IBM SPSS Statistics software, Version 28. All measures and manipulations in the study have been disclosed, alongside the method of determining the final sample size. No data collection was conducted following analysis of the data. Data for this study is available via the Figshare data repository and can be accessed at https://doi.org/10.6084/m9.figshare.25756800.v1. This study was not preregistered. Sample size was determined before any data analysis. A priori power analyses were conducted using G*Power to determine the minimum sample sizes required to test the study hypotheses. Results indicated the required sample sizes to achieve 90% power for detecting medium effects, with a significance criterion of $\alpha = 0.05$, were: $N = 108$ for the mixed between-within subjects ANOVAs and $N = 232$ for the one-way between groups ANOVAs. According to these recommendations, adequate statistical power was achieved. All univariate and multivariate assumptions were checked and found to be met. All scales and independent variables were normally distributed.

## Results

### Overview

The analysis of the current study including data screening processes, descriptive statistics, and hypothesis testing will be presented in this section. Hypothesis testing began with two separate mixed between-within subjects analysis of variance models (ANOVAs) to examine the impact of the experimental manipulation on the independent variables of body image and internalisation of appearance ideals and pressure. Finally, the effect of time spent using TikTok daily on restrictive and 'healthy' disordered eating behaviour was explored cross-sectionally using two separate one-way between-subjects ANOVAs.

### Data screening

Prior to statistical analysis, data were screened for entry errors and missing data. Of the 838 participants who initially consented to participate in the survey, 555 responses were insufficiently complete for data analysis. As participants were permitted to withdraw their consent by exiting the online survey, these results were excluded from all subsequent analyses. Of those that did not complete the study, the majority withdrew during the BISS (pre-test) and the ORTO-15, suggesting that these participants potentially experienced discomfort or distress

when asked to reflect on their appearance and their eating behaviours. Of the completed responses, nine were excluded due to not meeting the study's stated age eligibility and another case was excluded due to disclosure of a previous eating disorder diagnosis. The remaining data set comprised of 273 participants.

## Descriptive statistics

**Demographic characteristics.** In the current sample, 50% of participants reported being currently single and most participants (83%) were Caucasian, with 71% of participants indicating that they spent up to two hours per day using TikTok. Further demographic information is provided in Table 1.

**#ForYou: TikTok consumption demographics.** Participants in the current study reported that entertainment (75%), fashion (59%), beauty/skincare (54%), cooking/recipes (51%) and life hacks/advice (51%) content frequently occurred on their For You page. Largely in keeping with this, participants reported experiencing the most enjoyment from viewing entertainment (84%), life hacks/advice (57%), home renovation (56%), recipes/cooking (56%), and fashion (54%) content on their For You page.

In the current sample, 64% of participants reported being exposed to disordered eating content via their For You page. Only 15% of participants had not been exposed to any negative content themes. Further descriptive For You page content information is displayed below in Table 2.

Notably, 43% of the participant sample were frequently exposed to fitness and sports related content and the same percentage of the sample enjoyed seeing this content, suggesting that

**Table 1. Demographic characteristics of participants (N = 273).**

| Characteristic | *n* | % |
|---|---:|---:|
| Marital Status | | |
| Single | 138 | 50.5 |
| De-Facto | 119 | 43.6 |
| Married | 15 | 5.5 |
| Separated | 1 | .4 |
| Widowed | 0 | 0 |
| Ethnicity | | |
| Aboriginal | 9 | 3.3 |
| Torres Strait Islander | 1 | .4 |
| White | 227 | 83.2 |
| Hispanic, Latino, or Spanish origin | 2 | .7 |
| Asian | 18 | 6.6 |
| American Indian or Alaska Native | 1 | .4 |
| Middle Eastern or North African | 3 | 1.1 |
| Other | 12 | 4.4 |
| TikTok Daily Use | | |
| Less than 30-minutes | 37 | 13.6 |
| 30-minutes to 1 hour | 72 | 26.4 |
| 1 to 2 hours | 87 | 31.9 |
| 2 to 3 hours | 43 | 15.8 |
| 3 to 4 hours | 29 | 10.6 |
| 4 to 5 hours | 3 | 1.1 |
| Over 5 hours | 2 | .7 |

**Table 2. TikTok #ForYou content demographics (N = 273).**

| Content Theme | n | % |
|---|---|---|
| Most Frequently Viewed Content: | | |
| Entertainment | 206 | 75.5 |
| Fashion | 161 | 58.9 |
| Beauty/Skincare | 148 | 54.2 |
| Cooking/Recipes | 140 | 51.3 |
| Life Hacks/Advice | 140 | 51.3 |
| Animals | 135 | 49.5 |
| Dance | 134 | 49.1 |
| Fitness/Sports | 117 | 42.9 |
| Home Renovation/DIY | 111 | 40.7 |
| Education | 72 | 26.4 |
| Outdoors/Nature | 70 | 25.6 |
| Pranks | 69 | 25.3 |
| Art | 62 | 22.7 |
| Most Enjoyed Content: | | |
| Entertainment | 230 | 84.2 |
| Fashion | 148 | 54.2 |
| Beauty/Skincare | 147 | 53.8 |
| Cooking/Recipes | 152 | 55.7 |
| Life Hacks/Advice | 156 | 57.1 |
| Animals | 141 | 51.6 |
| Dance | 109 | 39.9 |
| Fitness/Sports | 116 | 42.5 |
| Home Renovation/DIY | 154 | 56.4 |
| Education | 101 | 37 |
| Outdoors/Nature | 92 | 33.7 |
| Pranks | 59 | 21.6 |
| Art | 76 | 27.8 |
| Negative TikTok Content: | | |
| Disordered Eating | 174 | 63.7 |
| Illegal Activity | 159 | 58.2 |
| Suicide | 151 | 55.3 |
| Bullying | 147 | 53.8 |
| Self-Harm | 123 | 45.1 |
| Violence | 108 | 39.6 |
| None of the above | 41 | 15 |

content broadly aligned with #fitspiration was consumed and appreciated by nearly half of participants. Concerningly, 40–60% of participants had been exposed to negative TikTok content via the For You Page, with content ranging from self-harm and suicidality to violence and illegal activity. No data was collected on the specifics of this content, however, and it is possible that some "negative" content may be framed from a proactive, preventative perspective, and this warrants further consideration.

## Hypothesis testing: Cross-sectional analysis

**Hypothesis 1: Daily TikTok use and disordered eating behaviour.** To test the cross-sectional analysis of this study, two separate one-way between-groups ANOVAs were conducted

**Table 3. Means and standard deviations of 'healthy' disordered eating and restrictive disordered eating across groups of TikTok daily use.**

| Daily TikTok Use | *n* | 'Healthy' DE | | Restrictive DE | |
|---|---|---|---|---|---|
| | | *M* | *SD* | M | SD |
| Low | 109 | 10.61 | 2.43 | 14.76 | 14.77 |
| Moderate | 87 | 10.54 | 2.79 | 15.53 | 13.15 |
| High | 43 | 10.56 | 2.85 | 18.16 | 13.33 |
| Extreme | 34 | 11.09 | 2.71 | 19.09 | 14.31 |

to explore the impact of daily amount of TikTok use on 'healthy' disordered eating and restrictive disordered eating behaviour. This was necessary as time on TikTok was measured categorically. Participants were divided into four groups according to their average daily time spent using TikTok (Low use group: 1 hour or less; Moderate use group: 1–2 hours; High use group: 2–3 hours; Extreme use group: 3+ hours). Homogeneity of variance could be assumed for each ANOVA as indicated by non-significant Levene's Test Statistics.

There was no statistically significant difference at the p < .05 level in ORTO15 scores for the four TikTok usage groups: $F(3, 269) = .38$, $p = .78$, indicating that 'healthy' disordered eating did not significantly differ across women who use TikTok for different periods of time per day. The effect size, calculated using eta squared, was .004, which is considered small in Cohen's [68] terms. This small effect size is congruent with the non-significant finding.

The second ANOVA measuring differences among EAT-26 scores across the four TikTok usage groups also yielded a non-significant result: $F(3, 269) = 1.21$, $p = .31$. Eta squared was calculated as .01, representing a small effect size [68] consistent with this non-significant result. The means and standard deviations of the four TikTok usage groups across dependent variables of 'healthy' and restrictive disordered eating, as measured by the ORTO15 and the EAT-26 respectively, are displayed in Table 3.

## Hypothesis testing: Experimental analyses

**Hypothesis 2: Body image satisfaction across groups from pre-test to post-test.** To evaluate the effect of the experimental intervention on body image, a 2 x 2 mixed between-within subjects ANOVA was conducted with condition (experimental vs control) as the between subjects factor and time (pre-manipulation vs post-manipulation) as the within subjects factor. All assumptions were upheld, including homogeneity of variance-covariance as indicated by Box's $M$ ($p > .001$) and Levene's ($p > .05$) tests [64].

The interaction between condition and time was significant, Wilks' Lambda = .98, $F(1, 271) = 6.83$, $p = .009$, partial eta squared = .03, demonstrating that the change in body image scores from pre-manipulation to post-manipulation was significantly different for the two groups. The body image satisfaction scores for women in both conditions decreased from pre-manipulation to post-manipulation. As anticipated, participants in the experimental condition reported a greater decrease in body image satisfaction than women in the control condition (see Table 4). This interaction effect is displayed in Fig 1.

**Table 4. Body image state scale scores for the control and experimental conditions before and after experimental manipulation.**

| Time Period | Control | | | Experimental | | | Total | | |
|---|---|---|---|---|---|---|---|---|---|
| | *n* | *M* | *SD* | *n* | *M* | *SD* | *n* | *M* | *SD* |
| Pre-manipulation | 147 | 4.39 | 1.74 | 126 | 4.10 | 1.58 | 273 | 4.26 | 1.67 |
| Post-manipulation | 147 | 4.22 | 1.81 | 126 | 3.66 | 1.71 | 273 | 3.96 | 1.78 |

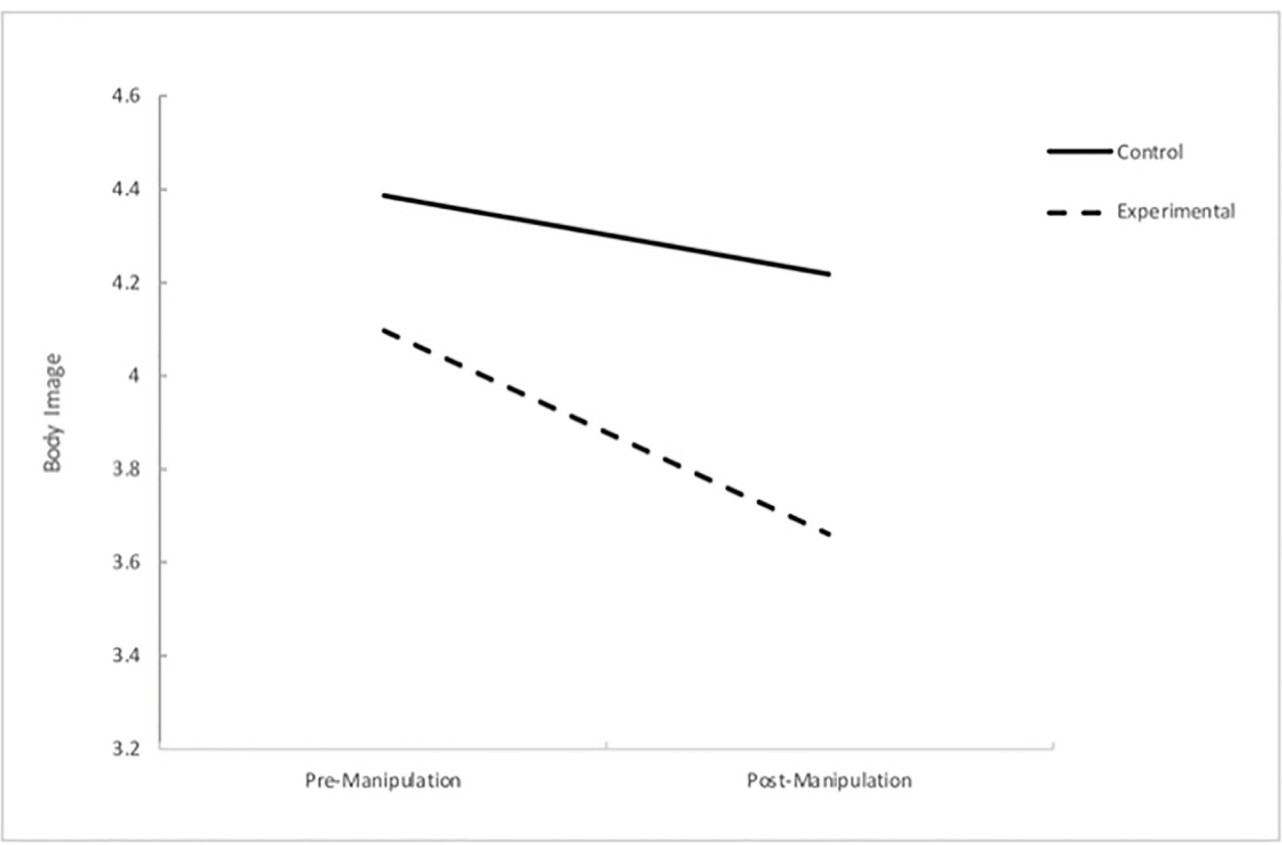

**Fig 1. Interaction effect of time and condition on body image scores.**

Although not consequential to the testing of the experimental manipulation, statistically significant main effects were also found for time, Wilks' Lambda = .89, $F$ (1, 271) = 32.99, $p = < .001$, partial eta squared = .109 and condition, $F$ (1, 271) = 4.42, $p = .036$, partial eta squared = .016. The means and standard deviations of these main effects are displayed in Table 4.

**Hypothesis 3: Internalisation of societal beauty standards across groups from pre-test to post-test.** A second 2 x 2 mixed between-within subjects ANOVA was conducted to investigate the effect of the experimental manipulation on participants' internalisation scores. All assumptions for the mixed model ANOVA were met with no violations.

A statistically significant interaction was found between group condition and time, Wilks' Lambda = .97, $F$ (1, 271) = 8.16, $p = .005$, partial eta squared = .029. This significant interaction highlights that the change in degree of internalisation at pre-manipulation and post-manipulation is not the same for the two conditions. Interestingly, the internalisation scores for women in the control group decreased from pre-manipulation to post-manipulation, whilst as anticipated, internalisation scores for women in the experimental group increased following exposure to the manipulation (see Table 5). This interaction is displayed in Fig 2.

No statistically significant main effects were found for time, Wilks' Lambda = .987, $F$ (1, 271) = 3.59, $p = .059$, partial eta squared = .013 or condition, $F$ (1, 271) = 2.65, $p = .104$, partial eta squared = .010. The means and standard deviations of internalisation scores for each condition at pre-manipulation and post-manipulation are displayed below in Table 5.

**Table 5. Internalisation of control and experimental conditions before and after manipulation.**

| Time Period | Control | | | Experimental | | | Total | | |
|---|---|---|---|---|---|---|---|---|---|
| | *n* | *M* | *SD* | *n* | *M* | *SD* | *n* | *M* | *SD* |
| Pre-manipulation | 147 | 67.18 | 13.86 | 126 | 68.91 | 14.25 | 273 | 67.98 | 14.04 |
| Post-manipulation | 147 | 66.80 | 15.45 | 126 | 70.79 | 15.84 | 273 | 68.64 | 15.73 |

## Discussion

### Overview

The current study investigated the effect of TikTok content on women's body image satisfaction and degree of internalisation of appearance ideals, and whether greater TikTok use contributed to increased disordered eating behaviour. In support of the hypotheses, exposure to pro-ana TikTok content significantly decreased participants' body image satisfaction and increased participants' degree of internalisation of appearance ideals. The hypothesis that greater daily TikTok use would contribute to increased disordered eating behaviour was not supported, as no statistically significant differences in restrictive disordered eating or 'healthy' disordered eating were found between the low, moderate, high, and extreme daily TikTok use groups.

### Cross-sectional findings

**Daily TikTok use and disordered eating behaviour.** Contrary to expectations, differences among groups on measures of restrictive disordered eating and 'healthy' disordered

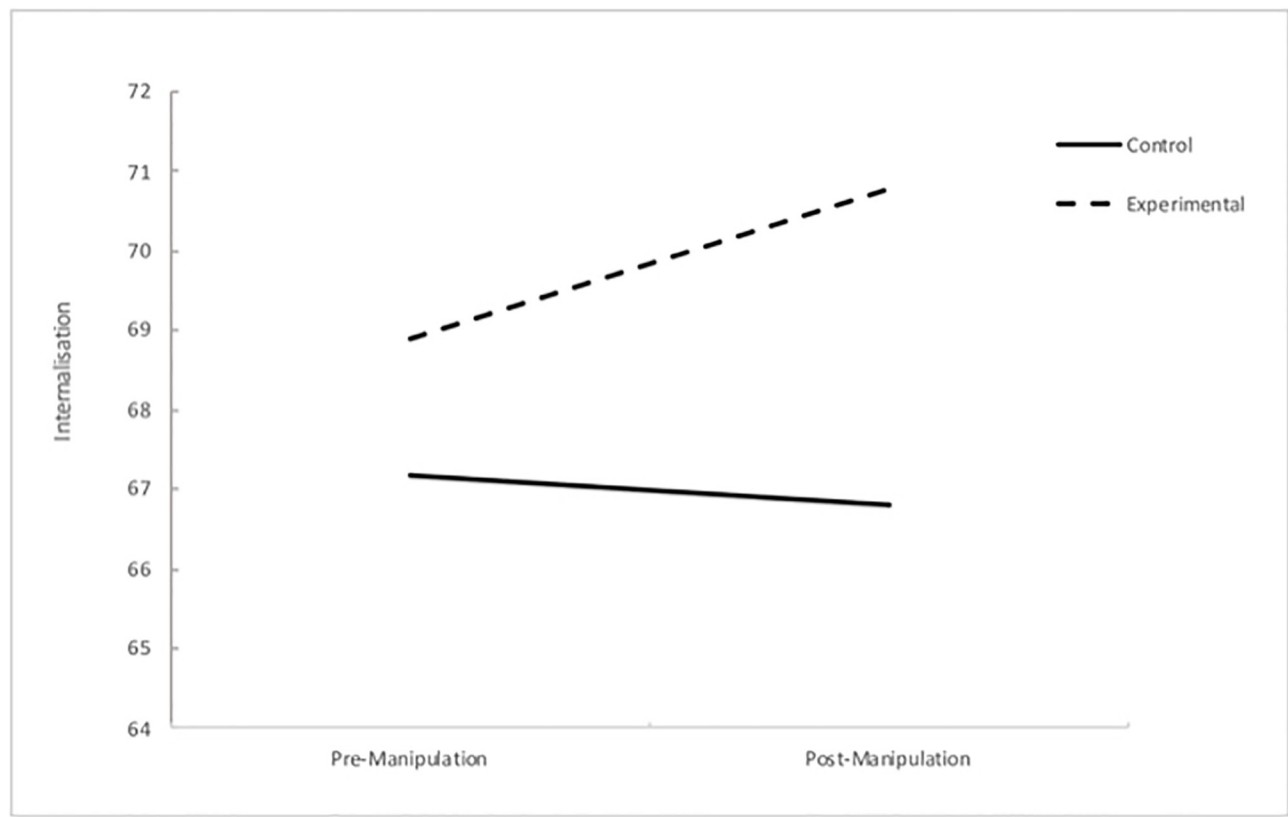

**Fig 2. Interaction effect of time and condition on internalisation scores.**

eating did not reach statistical significance. The proposed hypothesis that greater daily TikTok usage would be associated with disordered eating behaviour and attitudes was therefore unsupported. Despite lacking statistical support, participants categorised in the 'high' and 'extreme' daily TikTok use groups reported an average EAT-26 score of 18.16 and 19.09, respectively. Considering that an EAT-26 cut-off of $\geq$ 20 indicates potential clinical psychopathology, this mean score illustrates that exposure to TikTok content for two or more hours per day may contribute to a clinical degree of restrictive disordered eating.

The failure of the present study to detect any significant differences in disordered eating behaviours among participants with different TikTok daily usage does not align with the Transactional Model [33]. According to this model, risk factors such as low self-esteem and high thin ideal internalisation may predispose an individual to seek gratification via social media, resulting in body dissatisfaction and negative affect. The Transactional Model therefore proposes that a positive correlation exists between time spent on social media and body image dissatisfaction. Our findings also do not align with the conclusions Frieiro Padín et al. [34] drew from their review of the literature, in which a strong connection was identified between time on social media and heightened body image concerns and internalisation of the thin ideal, as well as eating disorder psychopathologies, though a distinction in outcome measures must be noted.

Based on the aforementioned sociocultural theory and previous research [see 28,43,48], it was assumed that increased body dissatisfaction as a result of increased time spent on social media (as stipulated by the Transactional Model), would lead to greater disordered eating behaviour. However, this was not supported statistically in the data. As postulated by Culbert et al. [69], disordered eating behaviour may instead only be a risk of media exposure if individuals are prone to endorse thin-ideals. Individuals in the present study that reported 'high' and 'extreme' daily TikTok use may have felt satisfied with their bodies and experienced lower thin-ideal internalisation. This could have potentially buffered the negative effect of greater TikTok content exposure and accounted for the lack of significant differences in disordered eating behaviour between groups. The quantity of TikTok consumption remains a pertinent question for disordered eating behaviour. As per the present study's brief experimental manipulation, findings suggest that high frequency of daily TikTok use does not necessarily contribute to greater disordered eating behaviour than short exposures to this content.

Content presented to the pro-ana TikTok group included a mix of explicit and implicit pro- eating disorder messages as well as fitspiration content. Fitspiration content presented in the current study included workout videos to achieve a "smaller waist" and "toned abs" where female creators with slim, toned physiques sporting activewear took viewers through a series of exercises, advising viewers that they would "see results in a week". In the present study, diet-related fitspiration content presented included the concoction of juices to "get rid of belly fat" and advice on the best "diet for a small waist" which requires avoidance of all meat, dairy, junk food, soda, and above all, to make "no excuses". Fitspiration style content in the current study totalled one-minute, compared to disordered eating themes which totalled six minutes. The integration of these various types of content, although reflective of the For You function in TikTok, impeded our ability to determine the singular impact of fitspiration or disordered eating content, respectively, on body image and internalisation of societal beauty standards, but did reflect social media as it is consumed beyond experimental research settings.

## Experimental findings

**TikTok and body image states.**    The hypothesis that women exposed to pro-ana TikTok content would experience a significant decrease in body image compared to women who

viewed the control TikTok content was supported. The present study found a significant interaction effect of body image between group condition (control vs experimental) and time period (pre-manipulation vs post-manipulation), as well as significant main effects. It is important to note that the statistic of interest in evaluating the success of the experimental manipulation is the interaction effect, thus main effects must be interpreted secondarily and with caution [64]. Women in the experimental group reported significantly lower body image satisfaction after exposure to the pro-ana TikTok content and compared to women who viewed the control content. This finding corroborates Festinger's [27] Social Comparison Theory that posits people naturally evaluate themselves in comparison to others. Exposure to the pro-ana TikTok content, consisting of various thin bodies and messaging around weight loss, may have provided the opportunity for women to engage in maladaptive upward social comparisons, resulting in reduced body image satisfaction. The present study upholds previous findings of Engeln-Maddox, Tiggemann, McComb and Mills, and Gibson [29,32,39,70] who suggest that visual exposure to thin bodies may adversely affect one's level of body image satisfaction and extends this research by replicating this finding in the context of a contemporary media platform, TikTok, and by utilising an experimental design.

Contradicting the present study and previous research, Pryde and Prichard [42] found no significant increase in young women's body dissatisfaction following exposure to fitspiration TikTok content. A potential explanation for this finding is that the performance of physical movements captured in fitspiration videos may shift the focus of viewers from aesthetics to functionality, highlighting physical competencies and capabilities which has been shown to improve body image satisfaction in young women [71]. Pryde and Prichard's [42] fitspiration content did not include typically occurring harmful themes as the present study did, potentially reducing the negative implications for body image satisfaction of exposure to such content in real world contexts.

Interestingly, women in the control group also reported a statistically significant decrease in body image satisfaction after viewing the neutral TikTok content, a finding that underscores the possible complexity of social media's influence on body image, as identified in research by Huülsing [72]. This is an unexpected finding, as the TikTok content displayed to the control group was selected specifically to be unrelated to appearance ideals and pressures. One possible reason for this result is the repetition of administration of the BISS within a short time period. Completing the BISS twice may have caused participants to focus more attention on their body appearance than usual, resulting in more critical appraisals regardless of the experimental stimuli to which they were exposed. This notion aligns with previous research that found focusing on the appearance of body was associated with lower body image satisfaction, whereas focus on the function of the body was associated with more positive body image states [71].

One potential explanation for this finding is that the control group stimuli was contaminated and produced an unintentional effect on body image scores. Two-minutes of footage within the seven-minute control group TikTok compilation presented the human body including legs, arms, and hands. Although this body-related content was neutral in nature, it may be that even 'harmless' representations of the human body are sufficient to elicit a social comparison response in participants or in some capacity, reinforce the #fitspiration motifs commonly depicted on TikTok [1], therefore impacting body image scores at post-manipulation. This possible explanation has implications for TikTok use and women's body image, as it suggests that viewing even benign content of human bodies for less than 10-minutes can have an immediate detrimental impact on body image states, even when this content is unrelated to body dissatisfaction, thinness, or weight loss. Furthermore, although a statistically significant

body image decrease was detected in the control group, this finding must be interpreted with caution due to the significant interaction effect obtained.

**TikTok and internalisation of societal beauty standards.** In accordance with the hypothesis, women in the experimental group reported a significant increase in their degree of internalisation of appearance ideals following exposure to pro-ana TikTok content. Women in the experimental group also reported significantly greater internalisation of appearance ideals than women in the control group. Conversely to the experimental group, internalisation scores of the control group decreased after viewing the neutral TikTok content. These findings are in line with the sociocultural theory, as women reported increased internalisation of societal beauty standards following exposure to media content explicitly and implicitly portraying the thinness ideal. The present study supports Mingoia et al's. [53] meta-analysis, which yielded a positive association between social networking site use and the extent of internalisation of the thin ideal and furthers this notion by replicating the finding with TikTok specifically and utilising an experimental design.

In the current study, participants were subject to a single brief exposure of pro-ana TikTok content, whereas most of the sample indicated that their TikTok use was up to two hours per day. This suggests that the degree of internalisation of appearance ideals in participants lives outside of the experiment are likely to be much greater. Mingoia et al. [53] also found that the use of appearance-related features on social networking sites, such as posting and viewing photos and videos, demonstrated a stronger relationship with the internalisation of the thin ideal than the use of social networking features that were not appearance-related, such as messaging and writing status updates. As TikTok is a video sharing app and most of its content generally features full-body-length camera shots rather than a face or head shot, this finding suggests that TikTok users could potentially internalise body-related societal standards to a greater extent than users of other social media apps that typically feature head shots.

The finding that women internalised societal beauty standards to a greater degree after being exposed to pro-ana TikTok content corroborates the sociocultural theory's emphasis of the significance of social influences in internalisation. TikTok users may be exposed to all three social influences (i.e., media, peers, and family) simultaneously on a single platform which may encourage internalisation of appearance-ideals in a more profound manner than any of these three influences in isolation. One point of difference between TikTok and other social media apps is that much content on the app is generated by "ordinary" individuals, rather than supermodels or celebrities. This enables blatantly insidious and diet-related content to circulate the app with less policing and scrutiny compared to content produced by an influencer or celebrity who may be more likely to be criticised or cancelled for socially irresponsible messaging and also provides the opportunity for more horizontal social comparisons and peer-to-peer style interactions rather than upward social comparisons.

Indeed, in their study of American teens, Mueller et al. [52] identified that girls were especially likely to engage in weight loss behaviour if a high proportion of girls with a similar BMI were also engaging in weight loss behaviours. This indicates that internalisation was strongest when appearance-ideals were promoted by alike peers. Due to the fact that much pro-ana TikTok content is created by young women, Mueller et al's. [52] finding has problematic implications for the young female users of TikTok, in that harmful diet-related messages could be internalised to a greater extent on TikTok than on other platforms and potentially lead to body image disturbances, disordered eating behaviour, and other negative outcomes among young women.

## General discussion

The findings of the current study are important but must also be understood within the broader context of participant's daily lives beyond their participation in this study. Everyday female-identifying individuals are exposed to a multitude of different sources of information from which body image related stimuli can be drawn. The present study's experiment was not conducted in a controlled environment due to its online nature, therefore researchers did not have the ability to assess and control for other pieces of body image-related information that participants might have consumed prior to participation that may have been salient for their body image. Further research is required to identify how sustained a change in body image states as measured by the BISS may be over time.

The findings of this study provide some insights into how social media influences disordered eating behaviour and mental health; a theoretical gap in the literature that Choukas-Bradley et al. [6] highlight as holding back research in this domain. In particular, the findings of the current study indicate that short periods of exposure to disordered TikTok content have an effect, while the high-range EAT-26 scores observed for those who engaged with TikTok for two or more hours a day also raise questions about the duration of exposure. Nonetheless, our findings demonstrate that short exposure periods are sufficient to have a negative effect on body image and internalisation of the thin ideal.

One point that may be readily overlooked in developing a theoretical framework around social media's influence is that the narrative arc of TikTok videos is such that users are exposed to many short stories in quick succession, which may have a different effect to longer form content from a single content creator. As Pierce [2] notes, the speed of exposure to overlapping, but separate narratives depicted in successive videos, is an important feature of TikTok content and may contribute to the influence of such platforms on disordered eating and body attitudes. Each piece of content serves as a standalone narrative but may also overlap and interact with the viewer's experience of the next video they watch to build a cumulative, normalised narrative of disordered body- and eating-practices.

In the current study, participants who engaged with TikTok for two-three hours a day were classified as high users, and those who used TikTok for three or more hours were classed as extreme. These rates of usage may, however, be quite normative, with Santarossa and Woodruff [73] citing three-four hours a day on social media as normative for their sample of young adults, though notably participants in the current study were only questioned about their TikTok usage, not their general use of social media.

While we examined the effect of pro-ana content in this study, that some changes were observed in the control group as well as the experimental group indicates that the social media environment, characterised as it is by idealisation, instant feedback, and readily available social comparison [6], may play a general role in diminishing positive body image attitudes and healthy aspirations. This is supported by Tiggemann and Slater's [35,36] research in which social media usage was found to correlate positively with higher levels of body image concerns, in contrast to time spent on the internet more generally, and this may be particularly true for visually oriented platforms that sensitize viewers to their own appearance and that of others. As noted previously, of the visually-oriented social media platforms, predominantly TikTok and Instagram, videos are commonly framed on TikTok so that the subject's whole body is visible, particularly in dance videos and in #GymTok content, where on Instagram, cameo style head-shot videos appear more likely to feature, which further suggests that TikTok may provide more body-related stimuli than other platforms, even when the intention of the content does not relate to body-image or #fitspiration.

Importantly, the algorithm on TikTok functions in such a way that those who actively seek out body positivity content may also be exposed to nefarious body-related content such as body checking, a competitive, self-surveillance type of content where users are encouraged to test out their weight by attempting to drink from a glass of water while their arm encircles another's waist. As McGuigan [74] reports, watching just one body checking video may result in hundreds more filtering through a user's For You page, with those actively attempting to seek out positive body image content likely to be inadvertently exposed to disordered content due to the configuration of the algorithm. This function of the For You page is demonstrated in the current study, with 64% of participants reporting having seen disordered eating content on their For You page, higher than any other kind of harmful content, including suicide and bullying. The current study did not assess participants' consumption of #FoodTok, #GymTok, and #Fitspiration. Engagement with these dimensions of TikTok and the type of content that participants seek out via the search function warrant consideration in future research.

The TikTok algorithm underscores Logrieco et al's. [18] findings that even anti-anorexia content can be problematic, especially given complexities in determining and controlling what is performatively problematic, including videos discussing recovery and positive body attitudes that may somewhat paradoxically further body policing and competition among users and consumers of social media content. Furthermore, as Logrieco et al. [18] highlight, TikTok is replete in both pro-ana and much more implicit body-related content that may be harmful to viewers, not to mention those creating the content, whose experiences also warrant consideration.

## Theoretical and practical implications

The present study bridged an important gap in the literature by utilising both experimental and cross-sectional designs to examine the influence of pro-ana TikTok content on users' body image satisfaction, internalisation of body ideals, and disordered eating behaviours. While the negative impact of social media on body image and eating behaviours has been established in relation to platforms such as Instagram and Twitter, TikTok's rapid emergence and unique algorithm warrant independent analysis.

The present findings have important theoretical implications for the understanding of sociocultural influences of orthorexia nervosa development. Notably, this study is one of the first to highlight the association between orthorexia nervosa and the tripartite model of disordered eating using an experimental design. The results illustrate that the internalisation of sociocultural appearance ideals predicts the development of 'healthy' disordered eating, as suggested by the tripartite theory. Western culture ideals do seem to influence the expression of orthorexic tendencies, thus caution should be exercised by women when interacting with appearance-related TikTok content.

Unlike explicit pro-ana content, which is open to condemnation, the moral and health-related discourses underpinning much body-related content in which thinness and health are espoused as goodness, reflects a new trend in diet culture masquerading as wellness culture [20,21]. Questions are raised around the ethics of social media algorithms when the technologically fostered link between recovery-focused content and disordered-content on TikTok is laid bare, particularly considering that extant research has found individuals with experience of eating disorders often seek out support, safety, and connection online [49] and in doing so on a platform like TikTok, may be exposed to more disordered eating content than the average user. Given visual social media platforms are associated with higher levels of dysfunction in relation to body image [4], the policy and ethics of such platforms warrant scrutiny from a variety of stakeholders in management, marketing, technology regulation, with psychology

playing an important role in the marketing of these platforms. As traditional journalistic platforms have been subjected to scrutiny and reform, so too must a climate of accountability be established within the social media nexus.

The widespread growth of social media may warrant greater concern than traditional forms of mass media, not only because of the full-time accessibility and diverse range of platforms, but also due to the prevalence of peer-to-peer interactions. According to the social comparison theory, comparison of oneself to others has traditionally considered more removed, higher status influences (e.g., celebrities, actors/actresses, supermodels) as a greater source of pressure than those in the individuals' natural environment (e.g., family and peers). Re-examination of this theoretical perspective is warranted considering the contemporary challenges of social media and the perpetuation of body image messages from alike peers. Furthermore, a diverse range of "content" may trigger disordered body- and eating-related attitudes, including #fitspiration and #GymTok, which poses challenges for social media platforms in regulating content. The inclusion of orthorexia in the milieu highlights the disordered nature of seemingly benign health practices and social media content.

That TikTok content containing explicit and implicit pro-ana themes may readily remain on the app uncensored exemplifies the importance of protective strategies to build resilience at the individual level. One such protective strategy is shifting focus from body appearance to functionality. Alleva and colleagues [71] investigated the *Expand Your Horizon* programme, designed to improve body image by training women to focus on body functionality. They report that women who engaged with the *Expand Your Horizon* programme experienced greater satisfaction with body image and functionality, body appreciation, and reduced self-objectification compared to women who did not engage with the program. Health professionals involved in the care of women with eating disorders and other mental health issues should also be educated to ensure they are knowledgeable about the social media content their clients may be exposed to, equipping them with skills to engage in conversations about the potential detrimental impacts of viewing pro-ana and other harmful TikTok content [53].

The administration of such programs in schools, universities, community groups, and clinical settings could prove effectual in the prevention of disordered eating and body image disturbance development and may reduce symptom severity of a pre-established disorder. Such programs must be developed with great care, however, given the propensity for even anti-anorexia content to have a negative effect on those consuming it [18]. The development of self-compassion may also build resilience in women, with research confirming that self-compassion can be effectively taught [75]. Subsequently, programs have been developed such as Compassion Focused Therapy (CFT) in which clients are trained to develop more compassionate self-talk during negative thought processes and to foster more constructive thought patterns [76]. The value of CFT has been established in the literature with both clinical and non-clinical samples and has promising outcomes particularly for those high in self-criticism [77].

Young women should be provided with media literacy tools that can assist in advancing critical evaluations of the online world. Digital manipulation of advertising and celebrity images is well known to many people, however, this awareness may be lacking regarding social media images, as they are generally disseminated within one's peer network rather than outside of it [33]. Media literacy interventions may educate women about how social media perpetuates appearance-ideals that are often unrealistic and unattainable [53]. As an example, Posavac et al. [78] revealed that a single media literacy intervention resulted in a reduction in women's social comparison to body ideals portrayed in the media.

Such interventions might be extended to female-identifying TikTok users to educate them on the manipulation of videos to produce idealised portrayals of the self. Media literacy should

be commenced from an early age by teaching children, adolescents, and adults to understand the influence of implicit messages conveyed through social media and to create media content that is responsible and psychologically safe for others [79]. Increased understanding of messages portrayed by social media content may prevent thin-ideal endorsement and internet misuse. Notably, however, the most effective approach would be to address the problem at its source and increase the regulation of social media companies, rather than upskill users in how to respond to harmful online environments, which creates further labour for the individual while allowing organisations to continue to produce harmful but easily monetizable content.

## Limitations and future directions

To meet the requirements to run multivariate analyses, the continuous data of body image and internalisation scores were dichotomised using a median split to create 'low' and 'high' groups for each variable. Although dichotomisation was necessary to perform appropriate analyses and power analyses deemed the sample size as adequate following performance of the median split, dichotomising these variables may have contributed to a loss of statistical power to detect true effects.

Limitations are implicated in the use of the ORTO-15 in the present study. The ORTO-15 does not account for different lifestyle factors that may alter a participants' response, such as dietary restrictions, food intolerances, or medical dietary guidelines. The discrepancies in literature surrounding the psychometric properties of the ORTO-15 may be attributable to the lack of established diagnostic criteria of orthorexia nervosa, cultural differences in expressions of eating disorders, and difficulty comparing research results in determining orthorexia nervosa diagnoses due to inconsistencies in testing questions and cut-off values [61]. Due to unacceptable reliability in the present study, a factor analysis was performed which identified a factor relating to health food preoccupation. This identified factor was used as the ORTO-15 measure and data from these 5-items were used in analyses and referred to throughout the present study as 'healthy' disordered eating. Using the 5-items related to 'healthy' disordered eating rather than the complete 15-item scale may not have accurately assessed participants' degree of orthorexic tendencies. Despite these limitations, the ORTO-15 is the only accepted measure of orthorexic tendencies available [63]. Additionally, more limitations would likely have been encountered by using the full 15-item measure lacking reliability, compared to utilising the 5-item factor with acceptable reliability.

Future studies of TikTok and disordered eating behaviour should incorporate a measure of social comparison to verify whether social comparison is the vehicle through which women experience decreased body image satisfaction after viewing TikTok content. Future research should also examine the influence of TikTok content creation on body image, internalisation of thinness, and disordered eating behaviour and explore the association between what individuals consume on TikTok and the social media content that they produce. This research should be conducted using more diverse samples of women, including transgender women, to determine whether the findings of the present study are relevant for this population given the unique challenges regarding body image and societal beauty standards that they may experience.

Longitudinal studies are also warranted to examine the effect of exposure to pro-ana TikTok content over time, and to assess the effects of pro-ana TikTok content on body image satisfaction and eating disorder symptomology over time. Further research on orthorexia nervosa is needed to establish a more reliable measure of orthorexic tendencies and this would enable future investigation of the impact of pro-ana TikTok content on the development of orthorexia nervosa, as well as individual differences as predisposing factors in the development of

orthorexic tendencies. Finally, future research should examine the efficacy of media literacy and self-compassion intervention programs as a protective factor specifically in the TikTok context, where disordered eating messages are more explicit in nature than traditional media and other social media platforms.

## Conclusion

The findings of the current study support the notion that pro-ana TikTok content decreases body image satisfaction and increases internalisation of societal beauty standards in young women. This research is timely given reliance on social media for social interaction, particularly for young adults. Our findings indicate that female-identifying TikTok users may experience psychological harm even when explicit pro-ana content is not sought out and even when their TikTok use is time-limited in nature. The findings of this study suggest cultural and organisation change is needed. There is a need for more stringent controls and regulations from TikTok in relation to pro-ana content as well as more subtle forms of disordered eating- and body-related content. Prohibiting or restricting access to pro-ana content on TikTok may reduce the development of disordered eating and the longevity and severity of established eating disorder symptomatology among young women in the TikTok community. There are current steps being taken to delete dangerous content, including blocking searches such as "#anorexia", however, there are various ways users circumvent these controls and further regulation is required. Unless effective controls are implemented within the platform to prevent the circulation of pro-ana content, female-identifying TikTok users may continue to experience immediate detrimental consequences for body image satisfaction, thin-ideal internalisation, and may experience an increased risk of developing disordered eating behaviours.

## Author Contributions

**Conceptualization:** Madison R. Blackburn, Rachel C. Hogg.

**Formal analysis:** Madison R. Blackburn.

**Investigation:** Madison R. Blackburn.

**Methodology:** Madison R. Blackburn, Rachel C. Hogg.

**Project administration:** Madison R. Blackburn, Rachel C. Hogg.

**Supervision:** Rachel C. Hogg.

**Writing – original draft:** Madison R. Blackburn, Rachel C. Hogg.

**Writing – review & editing:** Madison R. Blackburn, Rachel C. Hogg.

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
