## [Decision Letter · Decision Letter 0]

15 Feb 2024

PONE-D-23-33901#ForYou? The impact of pro-ana TikTok content on body image dissatisfaction and internalisation of societal beauty standardsPLOS ONE

Dear Dr. Hogg,

Thank you for submitting your manuscript to PLOS ONE. After careful consideration, we feel that it has merit but does not fully meet PLOS ONE’s publication criteria as it currently stands. Therefore, we invite you to submit a revised version of the manuscript that addresses the points raised during the review process.

 Please submit your revised manuscript by Mar 31 2024 11:59PM. If you will need more time than this to complete your revisions, please reply to this message or contact the journal office at plosone@plos.org. Please include the following items when submitting your revised manuscript:A rebuttal letter that responds to each point raised by the academic editor and reviewer(s). You should upload this letter as a separate file labeled 'Response to Reviewers'.A marked-up copy of your manuscript that highlights changes made to the original version. You should upload this as a separate file labeled 'Revised Manuscript with Track Changes'.An unmarked version of your revised paper without tracked changes. You should upload this as a separate file labeled 'Manuscript'.

We look forward to receiving your revised manuscript.

Kind regards,

Barbara Guidi

Academic Editor

PLOS ONE

Journal Requirements:

Additional Editor Comments:

The paper needs a MAJOR REVISION. Please improve the manuscript by following the weaknesses highlighted by the reviewers. Principally, the purpose and the motivation of the study should be clearly explained and the results need to be revised in depth.

Reviewers' comments:

Reviewer's Responses to Questions

**Comments to the Author**

1. Is the manuscript technically sound, and do the data support the conclusions?

Reviewer #1: Yes

Reviewer #2: Partly

2. Has the statistical analysis been performed appropriately and rigorously? 

Reviewer #1: Yes

Reviewer #2: I Don't Know

3. Have the authors made all data underlying the findings in their manuscript fully available?

Reviewer #1: No

Reviewer #2: Yes

4. Is the manuscript presented in an intelligible fashion and written in standard English?

Reviewer #1: Yes

Reviewer #2: Yes

5. Review Comments to the Author

Reviewer #1: This study is dedicated to the very important topic and is overall very thoroughly done. There are several things to revise:

- The purpose of this study is not clear - why additional study is needed on the topic where many studies have already been done (as the review in the paper shows)?  I agree that how social media influences body satisfaction and eating behavior is a very important topic, but making it explicit what this study added would certainly help readers

- The introduction (with review) feels too long and repetitive. I believe requires a clearer structure. I also recommend adding a theoretical scheme of the subject that you are studying. Currently it is not clear why some study is cited in one section, and not above where this topic was already raised.

- On page 4 TikTok is described as a platform where users have less autonomy over their newsfeed than on other platforms. I think you either need to revise this position or provide additional arguments. For example, Instagram and Twitter also have a page with recommended content which depends a lot on the content a user has interacted with previously.

- Terminology is not always consistent - for example on page 16 you say that you study "disordered eating behaviour" but on page 17 you say "restrictive disordered eating behaviour" (the latter is probably a part of the former but it is not self-evident)

- on page 8 you say that slim-thicc ideal may be more problematic than thin-ideal because in some countries who do not aspire to the thin ideal may be influenced more by the slim-thicc aesthetic. This argument is not clear to me. Yes, in *these* countries shim-thicc ideal might be more problematic, but not in other countries. Below that you say that slim-thicc ideal is less attainable because it requires surgeries  - this seems more logical

- on page 14 you cite a study about the influence of ethnicity on awareness and internalisation of ideas about thinness - not clear how why is it necessary, what argument it supports

- the link to the dataset provided only has a pdf with the tables and plots, not original data

Reviewer #2: Thank you for the possibility to review this manuscript. This manuscript aimed to explore the impact of pro-anorexia TikTok content on body image and internalization of beauty standards in a sample of 273 young women aged 18-28. The cross-sectional and experimental methods were used. Participants filled anonymous online questionnaire and were randomly selected to participate in a short experiment (watching video content). Next, participants were divided into four groups and variables of study were assessed. Pre- and post-measures (appearance satisfaction, internalization of appearance ideals and orthorexia eating attitudes) were compared after exposure to pro-ana content of TikTok video and neutral video for the control group. Results showed that women in experimental conditions experienced a decrease in body image satisfaction and an increase in internalization of stereotyped appearance ideals. Women in the control group also reported a decrease in appearance satisfaction. No changes in orthorexia attitudes were observed. The results of the study are important for public health, and health education since TikTok is very popular among young women worldwide. There are my comments that I believe will help to increase the quality of the manuscript:

Abstract.

1) I would appreciate it if the aim of the study were included.

2) The results of the comparison variables of the study in 4 groups based on average daily time on TikTok are missing in the abstract.

Introduction

1) I recommend rewriting the Introduction to make it 3-4 times shorter. Now, the length of this part is 14 pages.

2) I would appreciate it if the authors presented the main research question earlier in the Introduction and then presented the specific arguments for it.

3) The presentation of research should be more concentrated, generalization of the findings but not the presentation of separate research is recommended.

4) Presenting orthorexia, the authors are recommended to cite recent research on the state of the art of orthorexia research. Why is it important to assess orthorexia attitudes in the context of the present research?

5) The same comment is for the fitspiration content in social media. Why is fitspiration so specifically presented if it is not specifically analyzed in the research?

6) The clear aim of the study should be developed and presented, not only hypotheses.

7) The cross-sectional and experimental designs were used in the study, why do the authors mention only the experimental design presenting their research? Please clarify this issue in all parts of the manuscript.

Methods

1) Please clearly present the Procedure explaining and dividing two parts – cross-sectional and experimental. I would advise starting the section with Procedure and then presenting Participants.

2) The ethical aspects of the research are a big concern. It is obvious from the literature that the pro-ana content is damaging young women’s body image and well-being, however, the experiment with the damaging content (recorded video) was implemented. Is that ok from the ethical perspective? How was the possible damage compensated for study participants?

3) The questions related to the use of TikTok and its content should be presented in a detailed way (Tables 1 and 2). It is unclear how the information presented in Tables 1 and 2 was gathered. What exact questions were asked and what possible answers were provided. It is necessary for research that might want to replicate the research in future studies.

4) Please exclude information in lines 516-520 since it is presented further.

5) Is the Likert scale in EAT26 starting at 1 and ending at 6? Isn’t it 0 to 3? Please double-check the description of the instrument.

6) Why do the authors use the ORTO-15 instrument if it lacks psychometric characteristics? I recommend excluding it from research and using more sound instruments in future studies.

7) The separate Statistical Analysis section is missing (lines 658-664; 686-693).

Results

1) Please divide the Results section presenting the cross-sectional part of the study first, and the experimental study second.

2) Please exclude information presenting the sample from the Results part.

3) The results of Table 1 should be integrated into the text.

4) The name of the Table 2 is not appropriate. Also, it should be reconsidered if the results of the Table 2 might be presented in the text.

Discussion

1) Please clearly divide between the cross-sectional part and the experimental part. I miss the results for the lines 808-811; and 924-929.

2) Please do not provide new results in the Discussion.

3) The Discussion should also be rewritten and shortened focusing on the main research questions.

4) The conclusions seem too long and not specific. Please clearly state the conclusions from the cross-sectional part and the experiment. Please exclude specific practical applications from the Conclusions and write more general recommendation.

6. PLOS authors have the option to publish the peer review history of their article (what does this mean?). If published, this will include your full peer review and any attached files.

Reviewer #1: No

Reviewer #2: **Yes: **Rasa Jankauskiene

---

## [Author Response · Author response to Decision Letter 0]

6 May 2024

To whom it may concern,

We believe the updated version of the manuscript aligns with the style requirements of PLOS ONE. A detailed statement is included in the Procedure section of the Method concerning how consent from participants was obtained. The raw dataset is now available via the following link (also included in the body of the manuscript): https://doi.org/10.6084/m9.figshare.25756800.v1

We have provided responses to each item of feedback from the review panel in the document entitled, "Response to Reviewers" and have updated the manuscript accordingly. 

We look forward to hearing from you in due course regarding this updated version of the manuscript. 

Kindest,

Rachel

---

## [Decision Letter · Decision Letter 1]

9 Jul 2024

#ForYou? The impact of pro-ana TikTok content on body image dissatisfaction and internalisation of societal beauty standards

PONE-D-23-33901R1

Dear Dr. Hogg,

We’re pleased to inform you that your manuscript has been judged scientifically suitable for publication and will be formally accepted for publication once it meets all outstanding technical requirements.

Kind regards,

Barbara Guidi

Academic Editor

PLOS ONE

Additional Editor Comments (optional):

Reviewers' comments:

Reviewer's Responses to Questions

**Comments to the Author**

1. If the authors have adequately addressed your comments raised in a previous round of review and you feel that this manuscript is now acceptable for publication, you may indicate that here to bypass the “Comments to the Author” section, enter your conflict of interest statement in the “Confidential to Editor” section, and submit your "Accept" recommendation.

Reviewer #1: All comments have been addressed

2. Is the manuscript technically sound, and do the data support the conclusions?

Reviewer #1: Yes

3. Has the statistical analysis been performed appropriately and rigorously? 

Reviewer #1: Yes

4. Have the authors made all data underlying the findings in their manuscript fully available?

Reviewer #1: Yes

5. Is the manuscript presented in an intelligible fashion and written in standard English?

Reviewer #1: Yes

6. Review Comments to the Author

Reviewer #1: All my comments were sufficiently addressed by the authors and I don't have any additional comments. I think now the paper can be published. The data ideally should be in easier-to-read format (not .sav (SPSS), but e.g. csv), but it's not critical.

7. PLOS authors have the option to publish the peer review history of their article (what does this mean?). If published, this will include your full peer review and any attached files.

Reviewer #1: No

---

## [Editor Report · Acceptance letter]

16 Jul 2024

PONE-D-23-33901R1 

PLOS ONE

Dear Dr. Hogg, 

I'm pleased to inform you that your manuscript has been deemed suitable for publication in PLOS ONE. Congratulations! Your manuscript is now being handed over to our production team.

Kind regards, 

on behalf of

Dr. Barbara Guidi 

Academic Editor

PLOS ONE